# The Importance of HLA Assessment in “Off-the-Shelf” Allogeneic Mesenchymal Stem Cells Based-Therapies

**DOI:** 10.3390/ijms20225680

**Published:** 2019-11-13

**Authors:** Marta Kot, Monika Baj-Krzyworzeka, Rafał Szatanek, Aleksandra Musiał-Wysocka, Magdalena Suda-Szczurek, Marcin Majka

**Affiliations:** 1Department of Transplantation, Faculty of Medicine, Medical College, Jagiellonian University, Wielicka 265, 30-663 Kraków, Poland; marta.kot@uj.edu.pl (M.K.); aleksandra.musial@uj.edu.pl (A.M.-W.); magdalena.suda-szczurek@uj.edu.pl (M.S.-S.); 2Department of Clinical Immunology, Medical College, Jagiellonian University, Wielicka 265, 30-663 Kraków, Poland; mibaj@cyf-kr.edu.pl (M.B.-K.); rafal.szatanek@uj.edu.pl (R.S.)

**Keywords:** cell-based therapy, clinical trials, allogeneic, autologous, HLA, HLA-matching, immunomodulation, mesenchymal stem cells

## Abstract

The need for more effective therapies of chronic and acute diseases has led to the attempts of developing more adequate and less invasive treatment methods. Regenerative medicine relies mainly on the therapeutic potential of stem cells. Mesenchymal stem cells (MSCs), due to their immunosuppressive properties and tissue repair abilities, seem to be an ideal tool for cell-based therapies. Taking into account all available sources of MSCs, perinatal tissues become an attractive source of allogeneic MSCs. The allogeneic MSCs provide “off-the-shelf” cellular therapy, however, their allogenicity may be viewed as a limitation for their use. Moreover, some evidence suggests that MSCs are not as immune-privileged as it was previously reported. Therefore, understanding their interactions with the recipient’s immune system is crucial for their successful clinical application. In this review, we discuss both autologous and allogeneic application of MSCs, focusing on current approaches to allogeneic MSCs therapies, with a particular interest in the role of human leukocyte antigens (HLA) and HLA-matching in allogeneic MSCs transplantation. Importantly, the evidence from the currently completed and ongoing clinical trials demonstrates that allogeneic MSCs transplantation is safe and seems to cause no major side-effects to the patient. These findings strongly support the case for MSCs efficacy in treatment of a variety of diseases and their use as an “off-the-shelf” medical product.

## 1. Introduction

Regenerative medicine is currently a dynamically growing field of modern medicine. The use of different kinds of stem cells can be viewed as an alternative to organ transplantation and treatment of many diseases such as neurological or cardiovascular diseases [1,2] that cannot be effectively treated by conventional methods. The stem cell based therapies include embryonic (ESC) [3] and adult stem cells (adult SC) with the latter group composed of endothelial progenitor cells (EPC) [4], cardiac-derived progenitor cells (CDP) [5], cardiac stem cells (CSC) [6], and genetically reprogrammed, induced pluripotent stem cells (iPSC) [7]. Nonetheless, mesenchymal stem cells (MSCs) seem to be the most frequently used for this type of therapy. MSCs are relatively easy to isolate and expand in vitro. Moreover, they secrete cytokines and growth factors and have the ability to migrate to the site of an injury where they exert immunomodulatory and regenerative effects [8].

Among various sources of MSCs, perinatal tissues are of special interest in terms of their use in allogeneic transplantation. Birth-associated tissues including placenta, umbilical cord blood, amniotic fluid and amnion are widely available and can be used for therapeutic purposes [9,10,11,12,13,14]. Additionally, the acquisition of the birth-associated tissues does not require invasive surgery procedures, which becomes an advantage over other tissue sources such as bone marrow or adipose tissue. Although, bone marrow still remains the main source of MSCs for most preclinical and clinical studies [15,16,17,18,19,20], there has been a noticeable shift of interest towards other sources of these cells [21,22].

Numerous studies confirm that MSCs show a tremendous potential in the treatment of many diseases, including immune and non-immune ones. The results of hitherto studies have demonstrated several properties of MSCs that promote their beneficial effects, including, (i) ability to migrate to the site of injury, (ii) secretion of soluble factors, (iii) modulation of immune response, and (iv) ability to differentiate and transdifferentiate into various cell types. In vivo studies have revealed that MSCs promote angiogenesis, proliferation, and differentiation of progenitor cells. They also prevent fibrosis and apoptosis, and modulate immune responses [23,24,25,26]. Since tissue injury is always associated with an immune response, MSCs are recruited to a damaged tissue where they secrete a variety of factors including growth factors, cytokines, and chemokines [23]. Paracrine effect is now recognized as the primary mechanism by which MSCs promote tissue regeneration [24,27,28]. Other data also suggest that direct cell-to-cell contact and communication through gap junctions may be important in regenerative activity of MSCs [29].

It is fair to assume that immunological barriers accompanying allogeneic MSCs applications are similar to those governing solid organ and tissue transplantation. This review focuses on recent discoveries in the field of autologous and allogeneic stem cell transplants with special emphasis on MSCs-based clinical trials not only in the context of therapeutic properties of MSCs, but also of immunological hurdles in allogeneic cell therapies. We discuss immunomodulatory properties of MSCs and outline the importance of human leukocyte antigen-matching (HLA-matching) in MSCs transplantation. A better understanding of immunological interactions between the donor cells and the recipient will enable development of safe, effective, and personalized cell therapy based on allogenic MSCs.

## 2. Therapeutic Properties of MSCs

### 2.1. Immunomodulation—A Key Process in Tissue Regeneration

MSCs are considered to be hypoimmunogenic due to the lack of class II HLA expression. However, as previously described, class II HLA can be re-expressed under inflammatory circumstances [30]. The immunomodulatory activity of MSCs is demonstrated by their impact on T cells, natural killer T cells (NKT), B cells, dendritic cells (DCs), neutrophils, and M1/M2 macrophages [31,32]. In vitro and in vivo experiments and clinical trials showed that MSCs are able to modulate the immune system by suppressing immune responses (inhibiting proliferation and maturation of the immune cells) [30,32,33,34]. This mechanism has been described for both autologous and allogeneic MSCs [31,32,33,34]. The immunomodulatory effect of MSCs on the immune system can be mediated both through soluble factors and cell–cell interactions [35,36,37,38], however, the paracrine signaling pathways are considered as the key mechanisms by which MSCs influence other cells. The MSCs exhibit their immunosuppressive properties by secreting transforming growth factor-β1 (TGF-β1), prostaglandin E2 (PGE2), hepatocyte growth factor (HGF), indoleamine-pyrrole 2,3-dioxygenase (IDO), nitric oxide (NO), and interleukin-10 (IL-10) [38,39,40,41,42]. In fact, the immunomodulatory properties of MSCs are the sum of microenvironmental conditions of the tissue source that they have been isolated from [43,44,45].

Since MSCs express class I HLA and do not express class II HLA, they remain unnoticeable to the effector CD4^+^ T cells [31,46,47,48]. Interestingly, Yan et al. reported that Treg cells in MSCs co-culture showed higher immunosuppressive properties than in the absence of MSCs [30,49]. The studies have shown that MSCs are able to maintain T cells in a dormant state through the mechanism of Fas ligand/Fas receptor down-regulation at the cell surface [50,51]. Moreover, the transplanted MSCs have the ability to transform macrophages from pro-inflammatory M1 to anti-inflammatory M2 phenotype probably via IL-10 [43,52]. Recent studies have shown that macrophages in co-culture with MSCs increased phagocytic activity while the level of secreted inflammatory cytokines was decreased [53,54,55,56,57,58,59].

Nasef et al. reported the association between human leukocyte antigen-G (HLA-G), which is a non-classical human molecule class I protein, and the immunoregulatory function of MSCs [60]. HLA-G is an important factor that prevents rejection of the fetus by the mother’s immune system [57,61,62]. Secreted by MSCs, HLA-G mediates the induction of Treg lymphocyte proliferation [63,64], and may also exert a suppressive effect on allogeneic T cells proliferation. Secreted G-5 and G-7 isoforms have been shown to have a substantial impact on the allograft acceptance through the MSCs ability to keep the immune tolerance in check [65,66,67,68,69].

### 2.2. Secretory Activity of MSCs—Growth Factors and Extracellular Vesicles (EVs)

Mesenchymal stem cells represent a promising treatment approach not only because of their anti-inflammatory and immunomodulatory properties, but also because of their paracrine activity and the secretion of many factors [70]. The number of factors released by MSCs is remarkable. For example, secretion of proangiogenic factors such as vascular endothelial growth factor (VEGF), hepatocyte growth factor (HGF), and transforming growth factor-β1 (TGF-β1) is able to enhance in vivo angiogenesis through the increase of microvascular density thus promoting blood flow recovery in the ischemic tissue [27,71,72].

Different types of cells release small membrane vesicles, which are referred to as extracellular vesicles (EVs) [73]. They represent a modern therapeutic modality, where the cells themselves are not being used directly [74]. EVs may be carriers of therapeutic properties of MSCs. EVs were shown to transport DNAs, RNAs, miRNAs, proteins, and other important factors to target cells, acting as messengers in intercellular communication [75]. The interest in EVs is on the rise, especially because of their possible use in clinical settings [76]. Due to their immunomodulatory, regenerative or anti-cancer properties, EVs represent an intriguing approach in the treatment of cardiovascular, nervous, and immune system diseases. Certainly, the use of EVs might be an alternative to cell-based MSCs therapies, however, more data regarding their safety and therapeutic abilities in various disorders is needed to fully understand their potential [74,75,76].

## 3. The Essence of an Autologous and Allogeneic Stem Cells Therapies

Autologous transplantation involves isolation of own stem cells, which, after proper preparation, are transplanted back into the same patient. In an allogeneic transplantation, stem cells are collected from related or unrelated donors and transplanted into a selected recipient (Figure 1). Mismatches in HLA antigens between the donor and recipient are the most formidable immunological barrier to transplantation and result in serious complications such as engraftment failure, late rejection or graft versus host disease (GVHD).

Importantly, using cell therapy as a standard clinical treatment requires a safe and efficacious administration of the product at the optimal dosage. In the case of autologous therapy, the cells are derived from the patients that are usually burdened with comorbidities. Furthermore, the preparation of the autologous product usually requires more time to reach sufficient cell number, since the cells must be first of all isolated and then expanded before the actual administration. Moreover, the proliferation capacity of patient’s cells could be low due to an existing disease, premedication and age [17,18]. These pitfalls and the need for an immediate product availability prompted researchers to lean towards allogeneic stem cells to resolve these issues.

Stem cells isolated from allogeneic sources allow to acquire ready-to-use product in a relatively short time. The allogeneic stem cells are obtained from young, healthy donors and subjected to multiple quality control steps before their actual administration to the patient. In some diseases, time of transplantation is critical, thus, previously cryopreserved, readily available allogenic stem cells can be quickly expanded in sufficient quantities for administration. For these reasons, these cells have become very attractive as an ‘off-the-shelf’ therapeutic product.

Another issue that should be considered with regard to allogeneic therapy is the recipient’s immune response after transplantation. This response has been recognized in organ and hematopoietic transplants, which resulted in the use of immunosuppression to protect allograft from rejection [77]. MSCs are immune-privileged, have the ability to evade [78], and/or suppress [79] the immune system, therefore, non-matched MSCs are much better tolerated than other types of cells, which creates the opportunity of using them as an allograft without the need of concomitant to immunosuppression [80,81]. In fact, there are no reports of rejection or serious side effects after allogeneic MSCs therapies, which strongly supports the use of allogeneic MSCs as readily achievable (off-the-shelf), efficient, and safe treatment modality.

Although using allogeneic stem cells has obvious advantages there are reports suggesting that despite the immuno-privileged status of MSCs the immune response against these cells can still be initiated by the donor’s immune system [19,82,83,84,85,86,87,88,89].

## 4. Allogeneic and Autologous Stem Cell Transplant—Clinical Trials

In 1995, Lazarus et al. conducted the first clinical trials using bone marrow MSCs in non-Hodgkin’s lymphoma patients [90]. Since then, numerous clinical trials have been conducted involving both allogeneic and autologous MSCs sources (Figure 2A). According to the official database of the US National Institutes of Health, there are currently 750 clinical trials with MSCs at various clinical stages (www.clinicaltrials.gov; Figure 2A,B), out of which 203 studies have been completed. Out of 750 clinical trials, 315 (42%) constitute allogeneic and 435 (58%) autologous MSCs-based clinical trials. Over the last nineteen years, the number of clinical trials has increased significantly with a very sharp increase in the last ten years (Figure 2C). In 2009, there were approximately 66 registered clinical trials whereas in 2019, this number reached approximately 304, which constitutes approximately a 460% increase in the span of a decade. The MSCs are currently being investigated to evaluate their biomedical potential in treating numerous diseases (Figure 3). These include diseases associate with organs (bone, brain, heart, liver, and lung), immunity (autoimmunity, immune system diseases, and arthritis), digestive system (metabolic, digestive, and gastrointestinal diseases), and many more (Figure 3). Altogether these data show the versatility of MSCs-based therapy, a medical discipline that is currently on the rise.

Published results have shown that allogeneic MSCs can be safely administrated to humans without any relevant immune reactions [16,20,80,91,92]. In the first clinical study published by Hare et al. in 2009 [80], allogenic bone marrow-derived human MSCs (Prochymal) were tested in patients with myocardial infraction (MI). The trial provided evidence that this therapy was both effective and safe and resulted in significantly better ejection fraction in human MSCs-treated MI patients versus placebo. Another clinical study based on autologous and allogeneic bone marrow-derived MSCs involved patients with chronic ischemic cardiomyopathy (ICM) [16]. Both allogeneic and autogenic cells showed regenerative effects, and more importantly, patients who received allogeneic MSCs did not develop significant donor-specific alloimmune reactions in response to cell administration.

There are numerous examples of clinical trials in which excellent tolerance to allogeneic MSCs has been reported [16,17,18,19,78,92,93,94,95]. One of such clinical trials concerns a large group of patients with osteoarthritis where meta-analysis of 844 allogeneic MSCs transplantations concluded that this therapy is safe after a 21-month follow-up [96]. Similarly, no transplantation-related adverse events occurred in MSCs-treated patients with degenerative disc disease (DDD) [15,17,18,19]. The positive effects of allogeneic MSCs transplantations have been also reported in many randomized clinical trials, involving patients with lupus erythematosus [97], left ventricular dysfunction [16], ankylosing spondylitis [98], graft versus host disease, and other autoimmune diseases [99], such as Crohn’s disease [92]. Vega et al. [17] have performed a randomized multicenter study to assess the feasibility and safety of using allogeneic MSCs to treat chronic knee osteoarthritis. Their results strongly suggest that injection of allogeneic MSCs has a therapeutic effect and may be a valid treatment alternative for this disease. Other results suggest that MSCs due to their immunosuppressive properties and paracrine activity can facilitate a HSCs (hematopoietic stem cells) engraftment and lessen GVHD severity [100,101,102]. The MSCs administrations to patients with hematological diseases and GVHD provided a remarkable clinical response. Zhao et al. have monitored 47 patients with refractory acute GVHD out of which 28 patients, who received allogeneic, bone marrow derived MSCs, exhibited a reduced incidence and severity of GVHD [49].

It has been reported that other stem cell types may have similar properties to MSCs. It should be noted that cardiac-derived stem cells have immunomodulatory properties in vitro resembling those described for MSCs [103]. The ALLSTAR clinical trial concerning myocardial regeneration with the application of allogeneic heart stem cells (cardiospheres) showed that these cells expressed CD73, CD90, and CD105-classical markers characteristic for MSCs [104,105]. The use of cardiac stem cells as an allograft suggests that allogeneic cell therapy may be widely applicable.

Jansen et al. [106] performed a meta-analysis of preclinical data of stem cell therapy in ischemic heart disease. They reviewed data from 82 studies and concluded that MSCs were used in most of them. The analysis showed that: (1) both autologous and allogeneic cell therapy exhibit similar effects—similar improvement in left ventricular ejection fraction, and most importantly (2) there were no serious immune reactions reported in any of the studies, which suggested that this cell therapy was safe.

In general stem cells-based therapy is used in patients who do not respond to conventional and biological treatments [92]. Recent publications (2015–2019) have shown that bone marrow, adipose tissue, and umbilical cord are the most frequent sources of allogeneic MSCs used in clinical studies (PubMed, https://www.ncbi.nlm.nih.gov/pubmed). A large group of patients enrolled in MSCs-based clinical trials (about 20%) suffers from cardiovascular diseases—cardiomyopathy, myocardial infarction, heart failure (acute and chronic), or limbs ischemia (www.clinicaltrials.gov). Despite prevention and advanced treatment methods, morbidity and mortality are still very high. Numerous studies have confirmed that MSCs administrations to damaged myocardium resulted in significant improvement of myocardial contractility [2,16,20,79,95,107,108].

The MSCs based therapies also represent a new frontier in neurological disease treatments. The use of these cells in patients with neurological problems such as spinal cord injury (SCI), multiple sclerosis (MS), amyotrophic lateral sclerosis (ALS), or epilepsy turned out to be very beneficial. MSCs promote neurological improvement through the release of neuroprotective and neurotrophic factors and anti-inflammatory properties [109,110,111,112]. The positive outcomes of MSCs-based studies have translated, over the last few years, in a remarkable increase in the number of clinical trials where MSCs are being used in the treatment of neurological diseases (www.clinicaltrials.gov).

At our facility, an attempt to produce the “CardioCell” product from MSCs isolated from Wharton’s jelly (WJ-MSCs) has been implemented under the CIRCULATE project (Strategmed2/265761/10/NCBR/2015). The “CardioCell” is being administered to patients in three clinical trials: Acute Myocardial Infarction (AMI-Study, EudraCT Number: 2016-004662-25), Chronic Ischemic Heart Failure (CIHF-Study, EudraCT Number: 2016-004683-19), and Non-Option Critical Limb Ischemia (N-O CLI-Study, EudraCT Number: 2016-004684-40). Although, the project is still in progress, the results seem to point out to the “CardioCell” as a possible alternative in the treatment of cardiovascular diseases. Our in vitro and in vivo studies have shown so far, that WJ-MSCs possess high regenerative potential and most importantly they are safe for recipients [72,113]. Our initial clinical results suggest that the level of donor specific antibodies (DSAs) in patients does not change even after multiple administrations, which might indicate that immunization does not occur (unpublished data).

## 5. HLA Matching—Old and New Application Challenges

### A Brief Story about Human Leukocyte Antigens (HLA)

The antigen responsible for the rejection of allotransplant (mouse tumor grafting model) was described, for the first time, by Peter Gorer in 1936 [114]. Gorer’s work was further expanded by George Snell, who established H-2 locus encoding strong or major histocompatibility antigens capable of inducing quick graft rejection. Next, in 1958, Jean Dausset, Jon van Rod, and Rose Payne described antibodies in human sera that reacted with alloantigens on leukocytes, which were termed as the HLA (human leukocyte antigen) complex [115].

By the early 1980s, it became known that HLA genes are located on the short arm of chromosome number 6, and that they encode six different, very polymorphic series of determinants (A, B, C, DR, DQ, and DP) (Figure 4).

## 6. Genes and Proteins—a Blessing or a Curse

HLA are divided into three HLA classes: class I, II, and III. The first two are important for induction of adaptive immune response, [116,117]. Class I HLA proteins are divided into classical (HLA-A, -B, and -C) and non-classical (HLA-E, -F, and -G) molecules [118,119]. Classical class I HLA molecules are ubiquitous glycoproteins found on almost all nucleated cells [120,121,122] and platelets (limited expression possible) [123].

Expression of class II HLA molecules is limited to “professional antigen presenting” cells, such as dendritic cells, B lymphocytes, monocytes, and macrophages [116]. Activated T lymphocytes and renal microvascular endothelial cells may also express class II HLA [124,125]. The presence of a great number of allelic versions of the HLA gene is manifested by enormous HLA polymorphism (Figure 4), which plays a crucial role in antigen recognition.

Three mechanisms of allorecognition have been described so far: direct, indirect and semi-direct [126]. The direct pathway involves a mechanism by which the recipient T cells (CD8^+^) recognize donor’s HLA molecules and peptides presented by them without antigen processing by recipient cells. The direct T-cell allorecognition plays an important role in acute rejection [116]. The indirect pathway allows to recognize processed peptides of allogeneic histocompatibility antigens presented by self HLA molecules. Indirect response is dominated by CD4^+^ T cells and plays a role in chronic rejection [116]. Semi-direct recognition represents a cross-talk between the direct and indirect pathways observed in the context of transplantation, e.g., CD4^+^ T cells with indirect allospecificity can amplify or regulate direct allospecific CD8^+^ T cells [126].

Heterogeneity/diversity of HLA molecules and presented peptides enable an effective defense against a wide spectrum of intruders. This evolution strategy is guaranteed by a wide spectrum of different rows created by distal domains of HLA molecules as a consequence of high polymorphism of HLA genes (the most polymorphic in humans) [116]. On the other hand, alloantigens of grafted tissue are recognized by the same mechanism. If the HLA antigens differ between donor and recipient (and present different spectrum of peptides), the recipient’s immune system may recognize the graft as foreign and initiate an immune response resulting in graft damage. The level of HLA mismatches correlates with the strength of the immune system response. Based on this knowledge, HLA typing is used not only for matching a donor and a recipient but also to estimate the immunological risk of donor recognition.

## 7. Organ Transplantation—Focus on DSA

The overwhelming majority of literature indicates that HLA matching between the donor and the recipient is critical for graft (e.g., kidney, pancreas, skin, or heart) function and survival [127,128,129,130]. Since complete HLA match is usually very rare other strategies such as matching in HLA-A, -B, and -DR antigens only [131], matching in CREG (cross reactive groups) [132], acceptable mismatches program (defined by the lack of antibodies against donor’s HLA in the recipient serum), recipients desensitization, and finally, immunosuppression are being introduced [133]. HLA antigens have multiple epitopes, which can be recognized by specific antibodies (antigenicity) and can induce specific antibody response (immunogenicity). The epitopes are determined by polymorphic amino acid residues on the HLA molecule surface. Patches of closely located polymorphic residues (app. 3 angstroms radius) are called eplets [134] or functional epitopes [134]. Eplets are essential components of HLA molecules recognized by antibodies, which are major risk factors for graft failure.

Antibodies against HLA antigens may occur naturally or may develop as a result of previous grafting, blood transfusion, pregnancy, or common viral infection [135]. Antibodies specific for donors’ antigens (donor specific antibodies, DSA) may cause graft rejection. The presence of DSA before transplantation or DSA created de novo is an unfavorable indicator for graft survival especially if it leads to complement activation. The activation of complement cascade results in cell lysis/destruction. Even low titers of DSA may be associated with antibody-mediated graft rejection. Higher titers of DSA may pose an immediate risk, which may serve as the basis for the application of aggressive therapies allowing transplant survival. Therefore, it is necessary to identify anti-HLA antibody specificities as well as their biological activity for establishing acceptable mismatches. 

## 8. Hematopoietic Transplantation; from a Perfect Match to Haplocompatibility

Transplantation of hematopoietic cells is a standard procedure for many hematologic and genetic diseases (congenital immunodeficiency and inherited metabolic disorders) [116]. The highest probability of finding a well-matched donor is mostly possible between the members of the recipient’s family.

If a perfectly matched family donor is unavailable, a search for unrelated or related, mismatched donors is a procedure of choice. Unfavorable scenarios, which may occur after hematopoietic stem cell transplantations, include graft rejection (rare) and a graft-versus-host disease (GvHD).

DSA have been implicated in graft rejection in solid organ transplantation, but their role in hematopoietic transplantation seems to be also important [136,137,138]. Preformed DSA were associated with a high rate of graft rejection in patients undergoing haploidentical transplantation (match in one HLA haplotype), even if their level was low [139]. It was also observed that recipients of haploidentical grafts developed DSA de novo [140]. The mechanism of how DSA influence hematopoietic stem cell transplantation is still unclear.

Current knowledge about HLA strongly supports the fact that by overcoming the HLA barrier it is possible to increase the number of successful transplants and to prioritize acceptable, specific mismatches with a potentially lower immunogenicity.

## 9. Immuno-Privileged Status of MSCs

Determination of HLA is a necessity for organ, tissue, and hematopoietic stem cell transplantation. Donor-recipient HLA-mismatches can result in transplant rejection and GVHD [141]. The underlying reason for solid organ damage/rejection is the initiation of an adaptive response by the graft recipient’s immune system resulting in an allograft rejection [142]. A solid organ allograft is a vascularized structure composed of a great number of cells, which constitute a potential source of multiple alloantigens. These alloantigens can be utilized by the allograft recipient’s antigen presenting cells, mainly dendritic cells, to initiate the three allorecognition pathways (direct, indirect, and semi-direct) described above [143,144,145]. MSCs represent a different form of an allograft than solid organs. Although the comparison of these two forms of allografts poses many problems, it is, however, reasonable to assume that MSCs are not as structurally advanced as solid organs. They constitute a homogeneous population of cells with no vascularization, which together with the absence of class II HLA expression diminishes the allorecognition pathways activation in the allograft recipient. It should be also noted that the immunosuppressive properties of MSCs (release of TGF-β1, PGE2, IL-10, etc.) even further promote graft survival [38,39,40,41,42]. These and other facts clearly support the need of exploiting MSCs as an alternative to allograft transplantation.

The fact that most stem cells express low levels of class I HLA and lack class II HLA molecules has generated the presumption that stem cells are immune-privileged and do not cause an immunological conflict between the host and transplanted cells. In fact, most studies do not characterize allogeneic MSCs and their potential recipients in terms of HLA compatibility. The few published studies have demonstrated that adult HLA-mismatched MSCs and ESCs are not immune-privileged. In vivo studies with animal models have shown that allogeneic MHC-mismatched bone marrow MSCs may induce an immune response and subsequently become rejected. Intracardiac injection of allogenic MSCs in pigs has induced an immunological reaction [82]. The immune response has been reported at both cellular and humoral levels [82,83,84,85,86,87,88,89]. An immunological reaction was also observed in ESCs transplantation models. Allogeneic ESCs transplantation into injured myocardium in mice triggered cellular infiltration [146,147]. In general, human ESCs and MSCs do not express class II HLA, however, the expression of these molecules may significantly be upregulated during cell expansion in vitro, which may result in an immunological reaction after in vivo applications [148,149]. There are few factors that may stimulate MHC/HLA molecule expression on stem cells in vitro and in vivo: culture medium supplemented with growth factors (i.e., FGF) [149], oxygen conditions, or epigenetic modification in vivo [150]. These findings should alarm researchers to reevaluate the possibility of class II HLA upregulation during stem cell expansion designed for therapeutic purposes. Additionally, class II HLA can also appear after MSCs encounter the proinflammatory microenvironment of an injured tissue. It should be emphasized that in vitro conditions are different from those present in the source tissues where the stem cells interact with other cells, growth factors, cytokines, and/or extracellular matrix proteins. All of these factors can affect the HLA expression on these cells.

Data concerning patients’ sensitization after allogeneic MSCs transplantation are very limited. Current knowledge is mainly based on the results of four different clinical trials in which, in total, 90 patients have been studied [16,19,91,95]. These trials include the treatment of non-ischemic dilated cardiomyopathy—37 patients (POSIEDON-DCM trial) [95], advanced heart failure—30 patients [91], osteoarthritis—14 patients, and DDD—9 patients [19]). The level of sensitization in all of these patients was very low and was noticed only in a few percent of the studied patients after allogeneic MSCs injections. Since HLA typing of MSCs has not been determined before administration, it is possible that the level of detected antibodies correlated with HLA mismatches between the donor and recipient [95,151]. Negligibly low immunogenicity was also noted in other clinical studies. In the osteoarthritis and DDD clinical trials, the immune response was weak and transient. The donor specific anti-HLA antibodies (DSA) were detected in only two out of 13 patients included in the trial (knee osteoarthritis trial). It should be pointed out that in these studies, the recipients received only a single injection of MSCs, which may explain why only a few of them developed DSA [19]. There is a strong possibility that DSA levels may prove to be much higher in recipients who receive multiple allogenic MSCs injections, which only confirms the necessity of anti-HLA antibody monitoring in such therapies. Thus, donor/recipient HLA typing together with the assessment of DSA levels should be implemented for a better monitoring of the immune responses to MSCs.

## 10. Safety of MSCs-Based Therapy

The basic rule in clinical trials is to minimize side effects and maximize safety and efficiency of a therapy. The tissue source, multistage cell culture process involving isolation, expansion, cryopreservation, and culture conditions all introduce many variables, therefore standardization of all procedures and protocols is extremely important. The minimal criteria outlined by International Society for Cellular Therapy ISCT allow to obtain homogeneous MSCs populations [152]. In vitro expansion, conducted under controlled and previously validated conditions guarantees to obtain the required cell number of the highest quality. Heterogeneity of cell population accompanied with cellular senescence leads to unreliable clinical outcomes. It has been shown that senescent cells possess altered secretome profile and surface molecule expression, which handicaps their functionality [153,154]. The correct in vitro characterization of cells and reduction of manufacturing diversity are the most important challenges of MSCs-based therapies.

As shown in Figure 2A, allogeneic MSCs-based clinical trials constitute about 40% of MSCs transplantations. The vast majority of these studies do not include HLA matching and DSA monitoring. In these trials the enrolled patients are analyzed for the presence of IgG HLA class I antibodies before transplantation and a few months after transplantation without previously typing the HLA of MSCs donors [92]. In these cases, it is difficult to clearly state that MSCs do not cause immune response in patients. Nevertheless, as available data have shown, mesenchymal stem-cells treatment is well tolerated regardless of the source of the original cells [17,20,80,92].

Patient monitoring after cell administration is an indispensable part of any clinical trial. Long-term follow up allows the assessment of therapy’s safety. It should be noted that biological properties of mesenchymal stem cells derived from different sources are not the same [155]. The different type of MSCs used in the treatment of different diseases may generate different results, thus, the results obtained from one study cannot be extrapolated to another. Although the safety of MSCs therapy is generally accepted, treatment related adverse events cannot be excluded. A follow-up period in MSCs-based clinical trials has not been established so far, but usually is not shorter that a few months. For better safety assessment, in particular with regards to MSCs tumorigenic potential exclusion, some clinical trials are conducted with a two-year follow-up period [17,80,92,95,112].

## 11. Regulatory Issues for Clinical Trials in Humans

It is highly important to conduct clinical trials under unified regulations. Regardless of specificity of the study, clinical trials should comply with the guidelines of a good clinical practice, which are obligatory for advanced therapy medicinal products. The use of the advanced therapy investigational product (ATIMP) requires approval from competent authorities and ethic committees. The trials conducted in all EU countries are registered in the EudraCT database. Indisputable is the fact that informed consents must be obtained from every subject enrolled in the clinical trial. While planning clinical trials, the study risks are identified and minimized and potential benefits for patients are estimated. To maximize safety of the therapy, the occurrence of all adverse events should be reported and assessed by investigators [156].

Manufacturing of ATIMP should be carried out in accordance with good manufacturing practice (GMP). At the beginning of the manufacturing process a sponsor/manufacturer needs to receive a confirmation that a treatment based on biological materials (i.e., genetic material, cells, or tissues) meets all criteria for an ATMP set by EMA’s Committee for Advanced Therapies (CAT). Sponsors and manufacturers ensure the quality of the product by complying standard operating procedures and quality control system [157]. Donors should be previously screened for infectious diseases and other risk factors. In case of an ATIMP that contains human cells or tissues, the sponsor, manufacturer, investigator, and institution where the product is administrated should ensure that there is a traceability system to identify the donor/source of donation, to easily link the donation to the product and then to link the product to the subject [158].

The details about the manufacturing process and the reagents used are described in Investigational Medicinal Product Dossier (IMPD)—the main document assessed during clinical trial approval. All changes in ATIMP development should be clearly described and evaluated in terms of safety and efficacy of therapy. Duration of the follow-up period is determined by the sponsor depending on the type of ATIMP [159].

## 12. Conclusions—Future Issues for Consideration

Stem cell based therapies have provided a new therapeutic approach in the treatment of chronic diseases. The available data suggest that allogeneic MSCs therapy is safe mainly because it does not generate an immune response in the recipient after transplantation. These findings strongly support the need for the development of allogeneic MSCs-based therapies. Despite the presence of considerable evidence supporting the therapeutic potential of MSCs, the clinical implications in the case of HLA-mismatched MSCs are still unknown. In the light of a growing interest in allogeneic MSCs transplantations, a long term monitoring of the enrolled patients with regards to their immunological profiles are recommended. The analysis of recipients’ immunization status prior and after MSCs injection will provide the necessary evidence to solve debatable issues, and as a result transform MSCs therapy into an “off-the-shelf” treatment of many diseases.

## Figures and Tables

**Figure 1 ijms-20-05680-f001:**
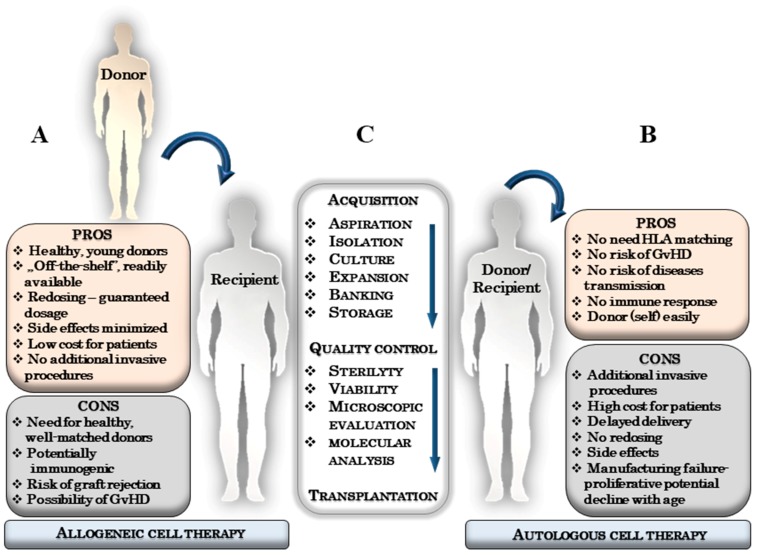
The advantages and disadvantages of allogeneic and autologous mesenchymal stem cells (MSCs)-based therapy. (**A**) Pros and cons of allogeneic cell therapy; (**B**) Pros and cons of autologous cell therapy; (**C**) Flow chart of MSCs-based therapy.

**Figure 2 ijms-20-05680-f002:**
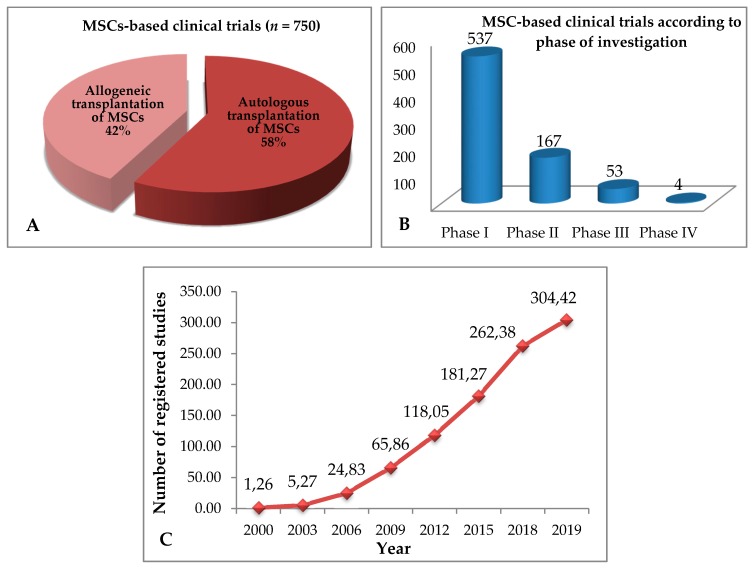
The number and percentage of MSCs-based clinical trials: (**A**) allogeneic versus autologous therapies. Autologous transplantations account for 58% of MSCs–based therapies; (**B**) Number of clinical trials according to the phase of investigation. (**C**) Number of registered clinical trials by year (data from www.clinicaltrials.gov).

**Figure 3 ijms-20-05680-f003:**
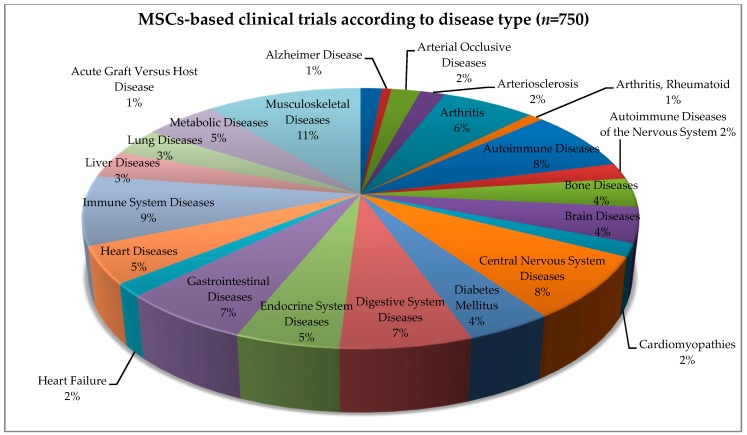
MSCs-based therapy in clinical trials associated with the treatment of different diseases. The pie chart shows the percentage of MSCs-based therapy with regards to a type of disease. (www.clinicaltrials.gov).

**Figure 4 ijms-20-05680-f004:**
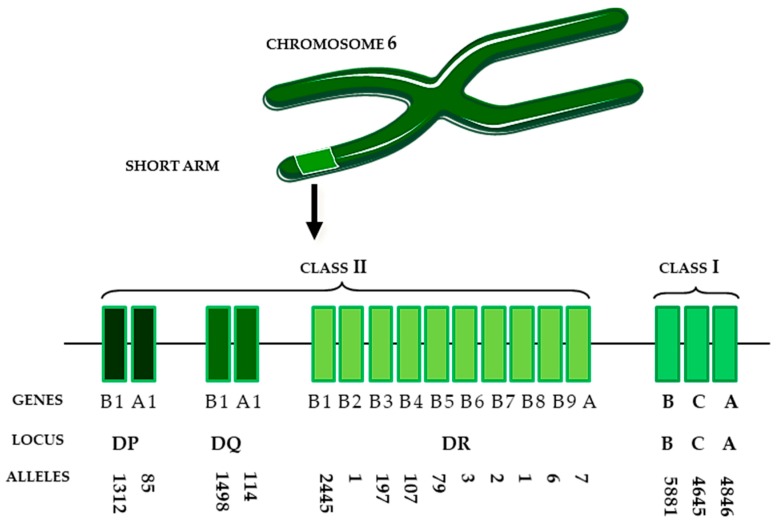
The human leukocyte antigen (HLA) region on chromosome 6. The ‘GENES’ row represents the designation of genes encoding alpha and beta chains associated with a corresponding HLA locus (DP, DQ, DR, B, C or A). The ‘ALLELES’ row represents the number of possible variants of a corresponding alpha or beta chain genes.

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
