# Peer review of "The Importance of HLA Assessment in “Off-the-Shelf” Allogeneic Mesenchymal Stem Cells Based-Therapies"

_ijms, 2019, doi:10.3390/ijms20225680_

Round 1
Reviewer 1 Report
The review by Kot et al highlighted the use of allogeneic MSC (predominantly) in various disease settings as well as their limitations.
Although this subject matter is of interest to many researchers the review in its current state needs some modification before it is can be published. I found the first half, parts 2-5, of the review to be long and unnecessary for the subject matter. Some of the concepts described could be included, in a summarised form and in relation to MSCs into the second part of the review. I understand that it is important to describe HLA matching but the review is not on this, nor DSA. The description of the semi-direct pathway is also very weak and no real explanation of its relevance following MSC transplantation.
The second half of the review was more relevant to their stated objectives in the introduction. The figures were also very nice. I would have liked a little more around the experimental evidence and the clinical trial data on MSC, as well as how they function. The last section on the 'problem' with MSC was also very relevant and I would have liked more info on this, in terms of the experiments undertaken to come to the conclusion that caution should be taken with allogeneic MSCs. The authors did not cover the concept of MSC EVs as a possible solution and given that there is a big push for these vesicles to be used clinically as an 'off the shelf' therapy I feel that the authors should have included some papers and discussion regarding this field.
Lastly I do not believe that the title reflects the content of the review and the authors may consider changing this to include MSC.
Minor points; some of the english language needs to be improved.
Author Response
Dear Reviewer,
Responses for Reviewer’s comments and suggestions:
Although this subject matter is of interest to many researchers the review in its current state needs some modification before it is can be published. I found the first half, parts 2-5, of the review to be long and unnecessary for the subject matter. Some of the concepts described could be included, in a summarised form and in relation to MSCs into the second part of the review. I understand that it is important to describe HLA matching but the review is not on this, nor DSA. The description of the semi-direct pathway is also very weak and no real explanation of its relevance following MSC transplantation.Response 1:
This part of manuscript has been shortened. All removed fragments are marked accordingly.
The second half of the review was more relevant to their stated objectives in the introduction. The figures were also very nice. I would have liked a little more around the experimental evidence and the clinical trial data on MSC, as well as how they function. The last section on the 'problem' with MSC was also very relevant and I would have liked more info on this, in terms of the experiments undertaken to come to the conclusion that caution should be taken with allogeneic MSCs. The authors did not cover the concept of MSC EVs as a possible solution and given that there is a big push for these vesicles to be used clinically as an 'off the shelf' therapy I feel that the authors should have included some papers and discussion regarding this field.Response 2:
We took under consideration Reviewer’s suggestions and added into manuscript proposed information and issues.
- we have cited more clinical data examples and experimental evidence in part: “Allogeneic and autologous stem cell transplant - clinical trials”
- we have added fragment describing therapeutic function of MSC (EVs, paracrine/secretory activity) in part : “Therapeutic properties of MSCs”
- we have discussed “problems” and cautions connected with allogeneic therapy as a new section: “Safety of MSCs-based therapy”
Lastly I do not believe that the title reflects the content of the review and the authors may consider changing this to include MSC.Response 3:
The title has been slightly changed. We have included Mesenchymal Stem Cells in the title.
Minor points; some of the English language needs to be improved.Response 4:
English has been improved by English native speaker.
Reviewer 2 Report
Marta Kot, Monika Baj-Krzyworzeka, Rafał Szatanek, Aleksandra Musiał-Wysocka and Marcin Majka reported a review manuscript entitled, “The Importance of HLA Assessment in “Off-the-Shelf” Allogeneic Cell Based-Therapies” to International Journal of Molecular Sciences.
Immunological barriers accompanying allogeneic MSCs applications are considered similar to those governing solid organ and tissue transplantation. The authors tried to focus on recent discoveries in the field of autologous and allogeneic stem cell transplants with special emphasis on MSCs-based clinical trials not only in the context of therapeutic properties of MSCs, but also of immunological hurdles in allogeneic cell therapies in this review. Also, the authors discuss immunomodulatory properties of MSCs and outline the importance of HLA-matching in MSCs transplantation. A better understanding of immunological interactions between the donor cells and the recipient will enable development of safe, effective and personalized cell therapy based on allogenic MSCs.
In conclusion, the authors state that stem-cell based therapies have provided a new therapeutic approach in the treatment of chronic diseases. The available data suggest that allogeneic MSCs therapy is safe mainly because it does not generate an immune response in the recipient after transplantation. These findings strongly support the need for the development of allogeneic MSCs-based therapies. Despite the presence of considerable evidence supporting the therapeutic potential of MSCs, the clinical implications in the case of HLA-mismatched MSCs are still unknown. In the light of a growing interest in allogeneic MSCs transplantations, a long-term monitoring of the enrolled patients with regards to their immunological profiles are recommended. The analysis of recipients’ immunization status prior and after MSCs injection will provide the necessary evidence to solve debatable issues, and as a result transform MSCs therapy into an “off-the-shelf” treatment of many diseases.
In this review, relatively there are few findings from the literature either in science or in clinical practice including on-going clinical trial.
It will be very important to include and discuss how different between solid organs MSCs in allogeneic transplantation both in scientific and of clinical application.
“A long tern monitoring” is also discussed on how stringent it would be in duration and safety/security.
Also, in practice of medicine, it is recommended to discuss the “regulatory” process and status in the authors’ and other countries or regions.
Author Response
Dear Reviewer,
Responses for Reviewer’s comments and suggestions:
Immunological barriers accompanying allogeneic MSCs applications are considered similar to those governing solid organ and tissue transplantation. The authors tried to focus on recent discoveries in the field of autologous and allogeneic stem cell transplants with special emphasis on MSCs-based clinical trials not only in the context of therapeutic properties of MSCs, but also of immunological hurdles in allogeneic cell therapies in this review. Also, the authors discuss immunomodulatory properties of MSCs and outline the importance of HLA-matching in MSCs transplantation. A better understanding of immunological interactions between the donor cells and the recipient will enable development of safe, effective and personalized cell therapy based on allogenic MSCs.
In conclusion, the authors state that stem-cell based therapies have provided a new therapeutic approach in the treatment of chronic diseases. The available data suggest that allogeneic MSCs therapy is safe mainly because it does not generate an immune response in the recipient after transplantation. These findings strongly support the need for the development of allogeneic MSCs-based therapies. Despite the presence of considerable evidence supporting the therapeutic potential of MSCs, the clinical implications in the case of HLA-mismatched MSCs are still unknown. In the light of a growing interest in allogeneic MSCs transplantations, a long-term monitoring of the enrolled patients with regards to their immunological profiles are recommended. The analysis of recipients’ immunization status prior and after MSCs injection will provide the necessary evidence to solve debatable issues, and as a result transform MSCs therapy into an “off-the-shelf” treatment of many diseases.
In this review, relatively there are few findings from the literature either in science or in clinical practice including on-going clinical trial.
Response 1:
We include more examples of clinical trials in section: “Allogeneic and autologous stem cell transplant - clinical trials”
It will be very important to include and discuss how different between solid organs MSCs in allogeneic transplantation both in scientific and of clinical application.Response 2:
We have discussed differences between solid organs and MSC transplantations and included this fragment in the section: “Immuno-privileged status of MSCs”
“A long tern monitoring” is also discussed on how stringent it would be in duration and safety/security.Response 3:
We have discussed this issue and we have emphasized the meaning of long term patients’ monitoring in terms of security of treatment.
Also, in practice of medicine, it is recommended to discuss the “regulatory” process and status in the authors’ and other countries or regions.Response 4:
We have added new section to manuscript “Regulatory issues for clinical trials in humans” in which we have explained some issues concerning regulation and registration of clinical product.
Round 2
Reviewer 1 Report
The manuscript resubmitted by Kot et al have addressed some of my concerns and they have included new sections which will be of interest to the reader. However the overall flow of the review is still missing and many of the sections seem to stand alone and read like they have been written by different authors. This really needs to be addressed before this review can be published. I am referring in particular to sections/parts 2 to 5. I understand why you are describing HLA and DSA and the 3 pathways of allorecognition but this needs to be linked to MSCs and their clinical use. Would it not be better to start with defining what MSC are (section 6) and how they function then moving onto how they are being used highlighting studies using autologous and allogeneic MSCs and then point out the 'danger' or not of allogeneic HLA? All the relevant information is mentioned in this manuscript I would just like more continuity.
Author Response
Dear Reviewer,
First of all, the authors wish to thank this Reviewer for the valuable comments regarding this review manuscript. Based on these comments the authors decided to reorganize the body of the review to achieve proper continuity of the manuscript. With that regard, the authors firstly decided to introduce the reader to the concept of mesenchymal cell therapy and then move to HLA, HLA mismatches and DSA. The last part of the review discusses the safety and clinical trial guidelines of allogeneic mesenchymal stem cell therapy, which is followed by concluding remarks.
Best regards,
Marta Kot
Reviewer 2 Report
Marta Kot, Monika Baj-Krzyworzeka, Rafał Szatanek, Aleksandra Musiał-Wysocka, Magdalena Suda-Szczurek and Marcin Majka reported a review manuscript entitled, “The Importance of HLA Assessment in “Off-the-Shelf” Allogeneic Mesenchymal Stem Cells Based-Therapies” to International Journal of Molecular Sciences.
Throughout the entire manuscript, it is not very clear why the authors state that “The importance of HLA assessment in “Off-the-Shelf” Allogeneic Mesenchymal Stem Cells Based-Therapies”.
Reviewing in the literature and comparing with “solid” organ transplantations, it is not very remarkably impacted on the difference of allogeneic HLA-mis-mach.
The authors are encouraged to state where such difference and stability occur in allogeneic mesenchymal stem cell transplantation other than secretory extracellular vesicles from mesenchymal stem cells.
Also, the facts and evidences how HLA mismatch affect negatively in allogeneic mesenchymal stem cell transplantation happens should be proposed and discussed.
Extensive revise is required before considering warrant for publication.
Author Response
Dear Reviewer,
First of all, the authors wish to thank this Reviewer for the valuable input regarding this review manuscript. Based on these comments the authors decided to restructure the body of the review to achieve proper continuity of the manuscript. With that regard, the authors firstly decided to introduce the reader to the concept of mesenchymal cell therapy and then move to HLA, HLA mismatches and DSA. The last part of the review discusses the safety and clinical trial guidelines of allogeneic mesenchymal stem cell therapy, which is followed by concluding remarks.
Below, please find the replies to this Reviewer’s comments:
Comment:
Throughout the entire manuscript, it is not very clear why the authors state that “The importance of HLA assessment in “Off-the-Shelf” Allogeneic Mesenchymal Stem Cells Based-Therapies”.
Reply:
The structure of manuscript has been restructured, thus proposed title become more adequate.
Comment:
Reviewing in the literature and comparing with “solid” organ transplantations, it is not very remarkably impacted on the difference of allogeneic HLA-mis-mach.
Reply:
The intention of this review was to make the reader aware of the potential complications associated with HLA mismatches in allogeneic mesenchymal stem cell therapy. Based on the presented data concerning mainly solid organ/hematopoietic cell transplants and a few reports (the authors are aware that there are not many available) discussing HLA mismatches in allogeneic mesenchymal stem cell therapy, the authors tried to highlight this problem. The authors strongly believe that proper HLA matching and antibody screening would only benefit an allogeneic stem cell therapy recipient.
Comment:
The authors are encouraged to state where such difference and stability occur in allogeneic mesenchymal stem cell transplantation other than secretory extracellular vesicles from mesenchymal stem cells.
Also, the facts and evidences how HLA mismatch affect negatively in allogeneic mesenchymal stem cell transplantation happens should be proposed and discussed.
Reply:
Although, this type of therapy is very promising there are a few reports stating that HLA matching between the recipient and the mesenchymal cell product might be of importance. It can be easily envisioned based on solid organ transplantation experience that the administration of a foreign product such as mesenchymal stem cells may cause immunization of the recipient. This could be especially the case when multiple administration of the allogeneic mesenchymal stem cell product is being delivered to the same recipient.
This review was meant to point out the necessity of antibody screening of the potential recipients before and during the whole mesenchymal stem cell treatment. The initial identification of antibodies present in the recipients could provide the assessment of possible risks associated with HLA mismatches. Continuing antibody screening after subsequent mesenchymal stem cell administration would allow monitoring of the levels of the initially identified antibodies and/or antibodies that developed de novo. All that valuable information could be then used for a more effective immunosuppression if the situation requires it.
It has to be stress out that this review, to the best of our knowledge, is the first to attempt to address the problems of HLA mismatches and DSA in allogeneic mesenchymal stem cell therapy. It has to be also mentioned that there are only scarce reports that tackle these issues and that it is very hard to draw legitimate conclusion based on this limited data. Nevertheless, the works discussed in the review signal the potential complications, which have to be taken into account for the benefit of the potential recipients of allogeneic mesenchymal stem cell therapy.
Best regards,
Marta Kot
Round 3
Reviewer 2 Report
Marta Kot, Monika Baj-Krzyworzeka, Rafał Szatanek, Aleksandra Musiał-Wysocka, Magdalena Suda-Szczurek and Marcin Majka reported a review manuscript entitled, “The Importance of HLA Assessment in “Off-the-Shelf” Allogeneic Mesenchymal Stem Cells Based-Therapies” to International Journal of Molecular Sciences.
This manuscript is intended to elucidate “The importance of HLA assessment in “Off-the-Shelf” Allogeneic Mesenchymal Stem Cells Based-Therapies” for the first time in DNA levels.
It is well written overall, and the content seems of impact to certain degree.
Clinical relevance in terms of allogeneic transplantation should be clarified.
If the HLA mismatch in allogeneic mesenchymal stem cells are different, how great would it affect to recipient?
In induction and sustaining the immune tolerance, the MSCs may be implicated for allogeneic organ transplantations, what would this immune tolerance and HLA mismatch in this situation be explained in clinical setting?
Figures 2b, 2c and 3, the data of the ratio of allogeneic or autologous to “total” in these distributions should be explored.
Author Response
Dear Reviewer,
Below, please find the replies to your comments:
Comments:
Clinical relevance in terms of allogeneic transplantation should be clarified.
If the HLA mismatch in allogeneic mesenchymal stem cells are different, how great would it affect to recipient?
In induction and sustaining the immune tolerance, the MSCs may be implicated for allogeneic organ transplantations, what would this immune tolerance and HLA mismatch in this situation be explained in clinical setting?
Reply:
All the currently available information concerning the above inquiries have been addressed in the ‘Immuno-privileged status of MSCs’. The potential consequences regarding HLA mismatches or DSA synthesis have been discussed in this section. The problem with MSCs-based therapy is that not many clinical trials even consider HLA typing and DSA monitoring to begin with, because this form of therapy is presumed to be safe. Indeed, most of the data support this clause, however, since very limited data is available on this issue, it cannot be said for certain that HLA typing and DSA monitoring can be disregarded. The authors understand this Reviewer’s need to explain how HLA mismatches and DSA monitoring impacts MSCs-based therapy, however, there is simply not enough data to draw concrete conclusions on this issue. The authors leave this as an open statement/question, which the potential reader should, in our opinion, consider when dealing with MSCs-based therapy.
Comment:
Figures 2b, 2c and 3, the data of the ratio of allogeneic or autologous to “total” in these distributions should be explored.
Reply:
Appropriate information has been inserted (highlighted) into the text. Please, see the ‘Allogeneic and autologous stem cell transplant - clinical trials’ section of the manuscript.
Best regards,
Marta Kot